# Spatiotemporal Fusion Prediction of Sea Surface Temperatures Based on the Graph Convolutional Neural and Long Short-Term Memory Networks

**Jingjing Liu** [1] , **Lei Wang** [2] , **Fengjun Hu** [1] , **Ping Xu** [1] **and Denghui Zhang** [1,*]

1 College of Information Science & Technology, Zhejiang Shuren University, Hangzhou 310015, China; liujingjing@zjsru.edu.cn (J.L.); hfj_zju@yeah.net (F.H.); xp_zju@yeah.net (P.X.)
2 Department of Marine Information Technology, East Sea Information Center SOA China, Shanghai 200136, China; lw_zju@yeah.net
* Correspondence: zdh_zju@yeah.net

**Abstract:** Sea surface temperature (SST) prediction plays an important role in scientific research, environmental protection, and other marine-related fields. However, most of the current prediction methods are not effective enough to utilize the spatial correlation of SSTs, which limits the improvement of SST prediction accuracy. Therefore, this paper first explores spatial correlation mining methods, including regular boundary division, convolutional sliding translation, and clustering neural networks. Then, spatial correlation mining through a graph convolutional neural network (GCN) is proposed, which solves the problem of the dependency on regular Euclidian space and the lack of spatial correlation around the boundary of groups for the above three methods. Based on that, this paper combines the spatial advantages of the GCN and the temporal advantages of the long short-term memory network (LSTM) and proposes a spatiotemporal fusion model (GCN-LSTM) for SST prediction. The proposed model can capture SST features in both the spatial and temporal dimensions more effectively and complete the SST prediction by spatiotemporal fusion. The experiments prove that the proposed model greatly improves the prediction accuracy and is an effective model for SST prediction.

**Keywords:** SST prediction; spatial correlation; GCN; spatiotemporal fusion





## 1. Introduction

Sea surface temperature (SST) changes have significant impacts on marine biology [1,2], the global climate [3,4], and extreme weather events [5], such as decreasing marine species diversity, altering global wind patterns, and increasing the incidence of floods. It also creates challenges for the ecosystem and for human life [6]. Through SST prediction, we can effectively understand ocean dynamics, which will enable us to adequately prepare for such challenges. However, the existing numerical prediction models require a profound understanding of and the ability to replicate the physical evolution of SST [7] to build a complicated model; these rely heavily on the accuracy of initial parameters, and it is difficult to capture the complex physical evolution accurately, which limits the development and accuracy of SST prediction. Along with the continuous advancement of deep learning technology, the data-driven modeling strategy has emerged as a powerful complement to numerical prediction models. It can learn and capture the patterns and features from a large amount of SST data without knowing the physical principles and identify the optimized parameters and weights for models in an iterative way. Thus, this strategy overcomes the limitation of numerical prediction models and has been effectively propelling the development of SST prediction technology. Therefore, deep learning-based SST prediction occupies an important position in the field of marine science.

Gaussian processes [8,9], support vector machines [10,11], genetic algorithms [12,13], and other machine learning algorithms have been commonly adopted for predicting SSTs. As machine learning, artificial neural networks, and deep learning techniques continue to evolve, these methods, models, and technologies have become widely used in the research on SST prediction [14]. However, the advantages of neural networks and deep learning technologies can only be fully utilized for SST prediction when there are sufficient amounts of data. With the rapid development of remote sensing technology [15,16], a large amount of SST data has been collected and stored. So, it is possible to more accurately and comprehensively mine rules from these SST data, which will result in better accuracy of SST prediction. Therefore, in recent years, prediction methods based on deep learning have rapidly developed and become the main research field of SST prediction. Artificial neural network models such as the feedforward neural network [17,18], which only has forward propagation with fully connected layers; the long short-term memory network (LSTM) [19,20], which can better process long time-series data by introducing a gate control mechanism; gated recurrent units (GRUs) [21,22], which optimize LSTM by simplifying the gate control; and the convolutional neural network (CNN) [23,24], which is able to better capture spatial features; as well as deep learning models composed of these different neural networks, are becoming the popular approach for SST prediction.

Choi et al. [25] proposed a method for predicting high-temperature events of SSTs using the LSTM model on time-series data from the Korean Peninsula. Usharani et al. [26] introduced an improved loss function for the LSTM model and achieved a better accuracy of SST prediction for six different locations in the Indian Ocean. Zhang et al. [27] proposed a hybrid model combining LSTM with a fully connected layer to address the regression problem of SST time series. Xiao et al. [28] combined LSTM and the AdaBoost ensemble learning model to more accurately predict the short and mid-term daily SST in the East China Sea. Yang et al. [29] developed a two-layer deep learning model consisting of a fully connected LSTM layer and a CNN layer for SST prediction along the coast of China. Xu et al. [30] proposed an encoder-based LSTM architecture to extract temporal and spatial information on SSTs, utilizing feature transformation and a decoder to predict the SST, and this achieved better accuracy in SST prediction. Ali et al. [31] used a two-layer deep learning model composed of GRUs and fully connected layers to predict SST with data from the Korea Hydrographic and Oceanographic Agency. Xu et al. [32] introduced the regional convolution LSTM (RC-LSTM) model, which leveraged the spatial advantages of the CNN to improve the SST prediction accuracy. Hao et al. [33] predicted the SST for the South China Sea using a convolutional long short-term memory (ConvLSTM) network and analyzed the impact of different input lengths, prediction lengths, and the number of hidden units on prediction accuracy. Yu et al. [34] integrated GRUs with the CNN to propose the DGCnetwork (deep gated recurrent unit and convolutional network) model for SST prediction in the East and South China Seas and achieved better accuracy than the GRUs. Xie et al. [35] significantly improved the SST prediction accuracy by integrating an encoder–decoder with GRUs. Xu et al. [36] combined the improved version of the LSTM and MIM (memory in memory) models with variational mode decomposition to detect change patterns for the SST. Qiao et al. [37] utilized a three-dimensional CNN to capture spatial features and LSTM to extract temporal dependencies, and they leveraged the attention mechanism to assign weights to each time step of the LSTM model, which enhanced the SST prediction accuracy. Sun et al. [38] proposed a temporal graph neural network for SST prediction in the northwestern Pacific Ocean; this used LSTM to capture the temporal features and a graph neural network to capture the spatial features. Yang et al. [39] designed a hierarchical clustering generator to cluster SST patterns with similarities, using a graph convolutional neural network to learn spatial correlations among clusters and feeding these into an RNN for SST prediction.

However, most of the existing studies on SST prediction capture spatial features through spatial partitioning, convolutional neural networks (CNNs), or unsupervised clustering. The spatial partitioning and CNNs depend on regular Euclidean space. The

unsupervised clustering cannot deal well with the spatial correlation between spatial points around the boundary of different clusters. Because the correlation between spatial points in the marine space is irregular and complicated, these methods limit the correlation expression and reflection between spatial points and also impact the accuracy improvement of SST prediction. So, it is necessary to explore more reasonable methods for spatial correlation discovery and mining. Recently, the graph convolutional network (GCN), which has more powerful spatial processing capabilities, has been commonly adopted in many fields, such as document classification [40], unsupervised learning [41], and image classification [42]. In order to address the above problems, this paper proposes a spatiotemporal fusion method for SST prediction that builds a graph convolutional neural network (GCN), constructs the graph data structure of SST data as input for the GCN, and designs the spatiotemporal fusion model (GCN-LSTM) based on the GCN and LSTM models. This GCN-LSTM model combines the spatial advantages of the GCN and the temporal expertise of the LSTM, so it can more effectively capture the spatiotemporal SST features and further improve the accuracy of SST prediction.

## 2. Materials and Methods

This paper explores the spatial correlation discovery and mining of SST data from four aspects, which include the regular boundary division for spatial interference elimination, convolutional sliding translation for spatial feature focusing, the clustering neural network for spatial feature extraction, and the graph convolutional neural network (GCN). The GCN part will also build the graph convolutional neural network, construct the graph structure of SST data, and, on this basis, design a spatiotemporal fusion prediction model (GCN-LSTM) by combining the GCN and LSTM approaches to improve the accuracy of SST prediction.

### 2.1. Regular Boundary Division for Spatial Interference Elimination

The regular boundary division [43] is realized by grouping spatial points in the sea area with close latitude and longitude into the same group, which reduces the interference caused by big differences between different groups and thus improves the effect of SST prediction. Within a group, there will be different spatial points where a different division method is used. Next, we will analyze three different division methods, which include horizontal rectangular division, square division, and vertical rectangular division. For a given sea area, using the same latitude resolution and longitude resolution, all spatial points in the sea area will be the set used for the regular boundary division.

The first method is the horizontal rectangular division. The assumption is that the SST difference in the longitudinal dimension is smaller than the difference in the latitudinal dimension over the same distance, so a flatter shape is used for division. The second method is vertical rectangle division. The assumption of this method is that the SST difference in the longitudinal dimension is greater than the difference in the latitudinal dimension over the same distance, so the vertically flatter shape is selected for division. The third method is the square division. The square division method assumes that the differences in the longitudinal and latitudinal dimensions over the same distance are similar, so the square shape is used for division.

Different regular division methods will lead to different data differences and the degree of spatial interference elimination will also be different, which will bring different effects for training and prediction.

### 2.2. Convolutional Sliding Translation for Spatial Feature Focusing

The convolution operation of the convolutional neural network is similar to the division mode in Section 2.1, and no explicit division operation is required. The convolutional neural network will arrange the spatial information according to certain convolutional windows and integrate and refine the data in each window through the sliding translation

of the convolutional window, so as to further realize spatial feature focusing and mine more feature information to improve the accuracy of SST prediction.

The two core concepts of the convolutional neural network are the convolution kernel *K* and the size of the convolution step *S*, which include the size of the horizontal and vertical directions, defined as $K_h$, $K_v$, $S_h$, and $S_v$, respectively. The convolution kernel is the window of the convolution operation. The convolution step indicates how the convolution kernel window moves and the distance of each move. The shape of the SST data is set as $(P_h, P_v, X)$, where $P_h$ is the number of horizontal spatial points, $P_v$ is the number of vertical spatial points, and *X* is the size of dimension for the SST data. The convolutional operation of a convolutional neural network will operate on a two-dimensional plane $(P_h, P_v)$.

With the definition of the convolution kernel and step size, the size of the output in the horizontal and vertical directions after the convolution operation can be calculated. The equation for calculating the output size in the horizontal direction is Equation (1).

$$O_h = (P_h - K_h)/S_h + 1 \tag{1}$$

where $O_h$ is the size of the horizontal output dimension, $P_h$ is the number of horizontal spatial points in the selected sea area, $K_h$ is the size of the horizontal direction of the convolution kernel, and $S_h$ is the size of the horizontal convolution step. The equation for calculating the output size in the vertical direction is Equation (2).

$$O_v = (P_v - K_v)/S_v + 1 \tag{2}$$

Finally, the output shape of the SST data $(P_h, P_v, X)$ after passing through the convolutional neural network will be $(O_h, O_v, F)$, where *F* is the number of the convolution kernel. The convolutional neural network fully processes and explores the input data in the spatial dimension. The output of the convolutional neural network can be fed into the LSTM model for further training in the temporal dimension on the basis of the spatial feature extraction, which is a powerful supplement to the spatial dimension for LSTM. Thus, the accuracy of SST prediction will be improved.

### 2.3. Spatial Feature Extraction by the Clustering Neural Network

Although the regular boundary division and convolutional sliding translation can play certain roles in improving SST prediction, they still have limitations because they are only based on the spatial points near each other, while there will be similarities and similar rules among the spatial points far away from each other. The clustering neural network can solve this problem to a certain extent. It will analyze and mine the data of spatial points to extract their spatial features and divide the spatial points into different groups according to the captured rules and similarities. Clustering is an unsupervised training mode that does not need to provide labels, which is in line with the need to find a better method for mining spatial correlations.

Self-organizing mapping [44] (SOM) is a commonly used unsupervised neural network for clustering. Each input of the input layer maps to a node of the hidden layer, while the output neurons compete with each other to be activated, and the neurons generate the final result in a self-organizing way, so it is called self-organizing mapping.

The SOM network consists of four main parts: initialization, competition, cooperation, and adaptation. Given an *M*-dimensional input $X = \{x_i: i = 1, 2, \ldots, M\}$, the connection weight between the node *i* of the input layer and the neuron *j* of the computing layer can be expressed as $W_j = \{w_{ji}: j = 1, 2, \ldots, N\}$, where *N* is the number of neurons in the computational layer. The first initialization section initializes $W_j$ to a relatively small connection weight tensor randomly. The competition process will find the neuron that matches the input using the Euclidean distance discriminant function, which is Equation (3).

$$d_j(X) = \sum_{i=1}^{M} (x_i - w_{ji})^2 \tag{3}$$

The neuron whose weight tensor is closest to the input tensor will be chosen as the winner. Once one neuron is selected, the probability of its neighboring neuron being selected is greatly increased. By setting $I(X)$ as the index of the winning neuron, the topological neighbors of $I(X)$ can be identified by Equation (4).

$$T_{j,I(X)} = \exp\left(-S_{j,I(X)}^2 / 2\sigma^2\right) \tag{4}$$

where $S_{j,I(X)}$ represents the distance between neuron $j$ and the winning neuron $I(X)$, and the parameter $\sigma$ is the neighbor radius used to control the neighbor scope. The following adaptation process adjusts the weights of the winners and their topological neighbors. The weight adjustment equation is Equation (5).

$$\Delta w_{ij} = \eta(t) \cdot T_{j,I(X)}(t) \cdot (x_i - w_{ji}) \tag{5}$$

where $t$ represents an epoch, and $\eta(t)$ represents the learning rate of epoch $t$. The weight update of each epoch makes the weights of the winning neuron and its neighbors closer to the input tensor, and the process is repeated until convergence is achieved.

For spatial feature extraction by the clustering neural network, the SOM neural network is first used to cluster the spatial points of the SST data and to divide them into multiple groups. Compared with other points in the sea area, the SST data of spatial points within each group have stronger similarity.

Next, clustering results can be used as the spatial correlation to improve SST prediction, which can be implemented in two ways. Taking LSTM as an example, the first way is to input the clustering results as a new feature into the LSTM model with SST data. The second way is to use LSTM to train each group. The first approach focuses on providing the LSTM model with more spatial information in the spatiotemporal dimensions. The second method aims to reduce the influence of the spatial points with weak correlations, make the LSTM model focus on the points with strong correlations in the spatial dimension, and mine the values and rules of SST data in the temporal dimension so as to improve the SST prediction.

### 2.4. Graph Convolutional Neural Network

Although spatial feature extraction by the clustering neural network solves the problem of not being able to handle non-adjacent nodes, it still has obvious boundaries between groups, and the connections between nodes in different groups are ignored to a certain extent. Therefore, a new model is needed that not only has the advantages of spatial feature focusing and spatial feature extraction but that can also mine the information between nodes in remote or different groups. It will further improve the effect of spatial correlation and, finally, optimize and improve the accuracy of SST prediction.

As long as there is a strong connection between two nodes, an edge can be added. The graph convolutional neural network (GCN) divides the interconnected points into a group, so it does not require explicit division and also does not depend on the regular Euclidian space. In other words, it uses an edge to replace the concept of a group. Any two nodes with a strong connection will have an edge, so there is no explicit boundary between groups, which addresses the problem of the lack of spatial correlation around the boundary of groups. Therefore, the GCN is a more perfect mechanism for mining spatial correlations.

The graph data structure mainly includes nodes, node data, the adjacency matrix, and the degree matrix. The node represents a spatial point in the graph, the node data are the dataset of each node, the adjacency matrix is used to indicate whether there are edges between any given two nodes, and the degree matrix represents the number of edges of each node. Because of the equivalence between two different spatial points in the sea, the graph discussed in this paper is undirected. Next, we will show how the GCN is built through the graph data structure described above.

Given a graph $G$, with node number $N$ and node data $X \in R^{N \times M}$, where $M$ is the dimension of the node data, the adjacency matrix is $A \in R^{N \times N}$, and the degree matrix is $D \in R^{N \times N}$. Because the node of a graph does not have an edge to itself, the diagonal values

of the adjacency matrix *A* from the top left to the bottom right are zero. In the field of neural networks, the nodes themselves also play a crucial role, so each node should have an edge to itself. Therefore, in order to apply the graph data structure to the neural network, it is necessary to add the identity matrix $I_N$ to the adjacency matrix *A* to form a new adjacency matrix, and at the same time, the identity matrix $I_N$ should also be added to the degree matrix to form a new degree matrix. The new *A* and *D* can be expressed by Equation (6).

$$\tilde{A} = A + I_N, \quad \tilde{D} = D + I_N \tag{6}$$

If there is an edge between two nodes, it indicates that there is a correlation between them. According to the idea of convolution, the information of the nodes with correlation to a node can be merged into this node, so that there is more relevant information for the neural network to learn. Merging information from other nodes into the node itself can be obtained by multiplying $\tilde{A}$ by *X*, i.e., $\tilde{A}X$. Because the information from the associated nodes is added to the node, the data values of the node with a large degree may become large, while the data values of the node with a small degree are relatively small. The neural network is sensitive to the size differences in the input data, which may cause gradient explosion or gradient disappearance.

Thus, the sum operation can be replaced by an average operation, and the degree of a node represents the number of its associated nodes, so it can be converted to an average value via dividing it by the value of its degree. For the entire graph, the degree matrix can be used here to solve the problem. The average values of the nodes of the entire graph can be realized by left multiplying $\tilde{A}X$ by the inverse matrix $\tilde{D}^{-1}$ of the degree matrix. This can be represented by Equation (7).

$$X' = \tilde{D}^{-1}\tilde{A}X \tag{7}$$

As we can see from Equation (7), $\tilde{D}^{-1}\tilde{A}$ is the normalization for the rows of the adjacency matrix $\tilde{A}$ by dividing the value of each node in the row *i* by $\tilde{D}_{ii}$, and $\tilde{A}$ is a symmetric matrix, so it should have the same operation for column in order to obtain better results. This can be achieved by right multiplying $\tilde{A}$ by the inverse matrix $\tilde{D}^{-1}$, as shown in Equation (8).

$$X'' = \tilde{D}^{-1}\tilde{A}\tilde{D}^{-1}X \tag{8}$$

At this time, another problem emerges: for the element $\tilde{A}_{ij}$ of adjacency matrix $\tilde{A}$, it is normalized twice, that is, $\tilde{A}_{ij}$ is divided by $\tilde{D}_{ii}\tilde{D}_{jj}$, which will certainly affect the predictive effect. To solve this problem, we can change $\tilde{D}_{ii}\tilde{D}_{jj}$ to $\sqrt{\tilde{D}_{ii}}\sqrt{\tilde{D}_{jj}}$. In the matrix form, this changes $\tilde{D}^{-1}$ to $\tilde{D}^{-1/2}$. After the change, $\tilde{A}_{ij}$ will be only normalized once. So far, for the entire graph, how the neighbor information of each node is aggregated to itself can be represented by Equation (9).

$$X''' = \tilde{D}^{-1/2}\tilde{A}\tilde{D}^{-1/2}X \tag{9}$$

From the perspective of the graph convolutional neural network, for a hidden layer $H^l$, the feature transfer between the nodes of the layer $H^l$ can be realized through Equation (9), and the features of the neighbors of the nodes can be passed to the nodes themselves to realize the convolution operation. On the basis of Equation (9), adding an activation function and trainable weight for nonlinear transformation will allow the next hidden layer $H^{l+1}$ to be obtained. Therefore, the propagation rule for the hidden layer of the graph convolutional neural network is expressed as Equation (10).

$$H^{l+1} = \sigma(\tilde{D}^{-1/2}\tilde{A}\tilde{D}^{-1/2}H^lW^l) \tag{10}$$

where $\sigma$ is the activation function, and $W^l$ is the trainable weight parameter.

From the above analysis, it can be seen that the graph convolutional neural network can process irregular data shapes, that is, the graph data structure, and integrate the features of the connected nodes through the convolution operation to fully explore the rules in the spatial dimension, thus breaking the limitations of regular boundary division,

convolutional sliding translation, and clustering neural network. Therefore, the graph convolutional neural network is more suitable for training and prediction using SST data and is able to improve SST prediction.

### 2.5. Construction of the Graph Data Structure for SST Data

In order to use the graph convolutional neural network for training and SST prediction, it is necessary to form the SST data into a graph structure. As described in Section 2.4, the three most important components of the graph structure are the nodes, the node data, and the edges. The data part is the time-series data of SST. Next, the nodes and the edges between the nodes will be constructed.

The approach assumes that the selected sea area contains $P$ spatial points, and the time-series data include the data of $D$ days. These $P$ points are the nodes of the graph structure, and each node will contain SST time-series data. The data of the entire sea area and the schematic diagram represented by nodes with their data are shown in Figure 1. Taking one data feature as an example, if there are multiple data features, it is only required to change the data of each node from one dimension to multiple dimensions.

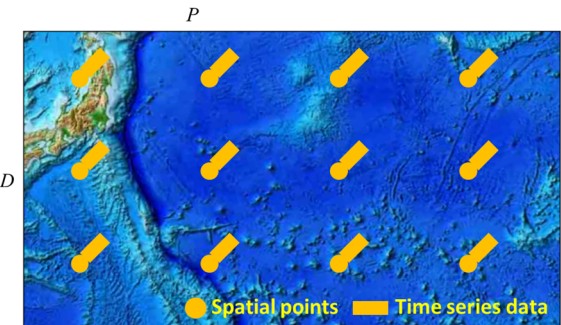

**Figure 1.** The schematic diagram of the SST data structure.

For the nodes in Figure 1, as long connections are added between the nodes, that is, the edges in the graph, it can transform the SST data into a graph data structure. In graph data structures, edges are represented by an adjacency matrix. Therefore, we need to find a way to generate the adjacency matrix. The purpose of this paper is to explore the law of spatial correlation and to integrate the information for the connected spatial points during model training and prediction so as to improve the prediction accuracy. In the field of SST data, it is to find the method used to identify nodes with a strong correlation relationship.

The easiest way is to calculate the correlation relationship using the distance between two spatial points. However, using the distance will have some limitations, because the time-series data of the spatial points are not considered, and the SST data of the two spatial points relatively far away may also have similar rules. Therefore, this paper determines whether there is an edge between the two points by defining a threshold and checking if the value of the correlation coefficient ($r$) is greater than it. The value of $r$ reflects the correlation of data between spatial points and can be used to judge the strength of the relationship between them. The definition of $r$ is shown as Equation (11).

$$r = \frac{\sum_{i=1}^{n} (x_i - \overline{x})(y_i - \overline{y})}{\sqrt{\sum_{i=1}^{n} (x_i - \overline{x})^2 \sum_{i=1}^{n} (y_i - \overline{y})^2}} \tag{11}$$

By using $r$ to determine the strength of the relationship between two nodes, the adjacency matrix $A$ of the graph can be expressed as Equation (12).

$$A_{ij} \in A, \quad A_{ij} = \begin{cases} 1, & r_{ij} \geq \alpha \\ 0, & r_{ij} < \alpha \end{cases} \tag{12}$$

where $r_{ij}$ is the value of $r$ between the $i_{th}$ and $j_{th}$ node, and $\alpha$ is the threshold of $r$ where there is an edge between two nodes. The range of $\alpha$ is (0, 1); in general, a value close to 1 will be taken. The adjacency matrix is calculated by Equation (12). The $r$ value between the node and itself is 1, and the parameter $\alpha$ is less than 1, so the adjacency matrix already contains the identity matrix. It is not required to add the identity matrix to $A$, and the graph convolutional neural network can directly use the adjacency matrix $A$.

An edge between spatial points in the sea area will be added through the adjacency matrix, and the two points with a strong correlation will establish a connection. In this way, the graph structure of SST data is formed through nodes and the adjacency matrix. Now, we have the graph data structure of the SST data, which can be expressed as Equation (13).

$$G = (X, A) \tag{13}$$

where $X$ is the node and data of the graph ($X \in R^{P \times D}$), and $A$ is the adjacency matrix of the graph ($A \in R^{P \times P}$). $P$ is the number of spatial points, and $D$ is the number of days of the time-series data. The graph structure will be used as the input for the graph convolutional neural network to realize the analysis and mining in spatial dimensions for SST and to improve the accuracy of SST prediction.

### *2.6. The Spatiotemporal Fusion Model for SST Prediction Based on the GCN and the LSTM*

It can be seen from Section 2.4 that the graph convolutional neural network (GCN) can fully implement feature extraction in the spatial dimension, but it has no special advantages for processing time-series data. LSTM is a deep learning model specially used to deal with time-series data. Therefore, combining the GCN and the LSTM can create a spatiotemporal fusion model, GCN-LSTM, for SST prediction, which will integrate the advantages of neural networks in the spatial and temporal dimensions.

In order to seamlessly integrate the GCN and LSTM models, we need to first adjust the SST graph data structure $G = (X, A)$ described in Section 2.5. The shape of the node and data $X$ is ($P$, $D$), and the shape of the adjacency matrix $A$ is ($P$, $P$). On this basis, the time step $T$ of the LSTM model is introduced, the shape of $X$ is adjusted to ($P$, $D$, $T$), and the shape of the adjacency matrix $A$ is unchanged. After the time step is added, each time step has a separate graph. The GCN will train all graphs corresponding to the time steps. If the size of the prediction window $F$ is taken into account as prediction results, the number of graphs will be generated based on $F$ after the training and prediction are completed. After adding time step, the diagram of the GCN is shown in Figure 2.

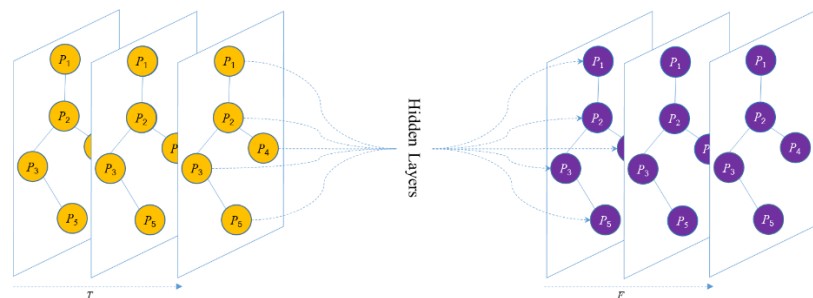

**Figure 2.** The diagram of the GCN after adding the time step.

As we can see from Figure 2, multiple SST graphs are trained through the hidden layers of the GCN and related nonlinear transformations, and the graphs, as the prediction results, will be generated based on the size of the prediction window $F$.

After the input data of the GCN is adjusted and the LSTM is added, the model becomes the proposed spatiotemporal fusion model for SST prediction, GCN-LSTM. The GCN-LSTM model is composed of four parts. The first part is data preprocessing and graph structure construction. The second part uses the GCN to train the SST graphs with time steps to realize spatial feature extraction. The third part feeds the training results of

the GCN into the LSTM for further time-series data processing. The last part generates the final prediction results through the fully connected layer. The structure of the GCN-LSTM model is shown in Figure 3.

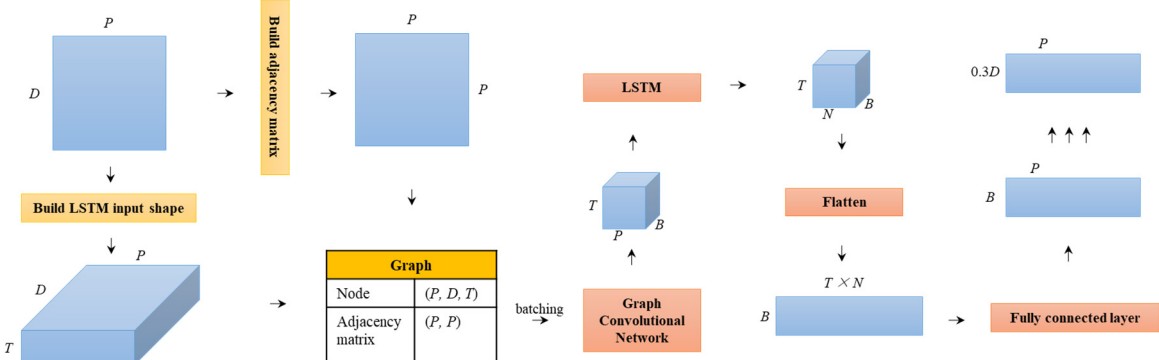

**Figure 3.** The structure of the GCN-LSTM model.

In Figure 3, $P$ is the number of nodes of the SST graph and the number of spatial points in the selected sea area, and $D$ is the number of days of the SST time series for each spatial point. The shape of the initial input data of the model is $(P, D)$. First, the adjacency matrix is constructed for the model according to Equation (12) in Section 2.5, and the shape of the adjacency matrix is $(P, P)$. Then, the time step of the LSTM is introduced to construct the input shape of the LSTM, and the shape of the input data is adjusted from $(P, D)$ to $(P, D, T)$ as nodes and node data of the graph structure. After the nodes, data, and adjacency matrix are obtained, the graph data structure of SST data is constructed by combining them. The shape of each batch of data is $(P, B, T)$, which is the input for the GCN. After convolutional and nonlinear transformations, the shape of the GCN output is the same as that of the input, so it is still $(P, B, T)$. It completes the feature extraction in the spatial dimension. Next, the graphs trained by the GCN will be input into the LSTM to conduct the feature analysis and extraction in the temporal dimension. The output shape is $(N, B, T)$, where $N$ is the number of hidden units in the LSTM. In order to generate the final prediction result of the graph, it is required to flatten $(N, B, T)$, reshape it to two dimensions, and adjust its shape to $(B, T \times N)$. Since the number of nodes in the final graph is $P$, another fully connected layer is needed here to further adjust the output shape to $(B, P)$. At this point, a batch of training and prediction data in the temporal dimension has been completed. The dataset is divided into 70% data as the training data and 30% data as the test data. Therefore, when all batches are trained and predicted, the final output result of the GCN-LSTM model is $(P, 0.3D)$. It represents the final graph as the prediction result, which contains $P$ nodes, and the size of the SST data for each node is $0.3D$ in the temporal dimension.

The GCN-LSTM model changes the input graph from one to multiple graphs according to the time step of the LSTM. For the graph of each time step, the feature extraction and mining in the spatial dimension are fully conducted through the spatial correlation identified by the GCN, so that the information between nodes with edges can be transferred and integrated with each other. Then, the LSTM is used to further mine the features of the SST data in the temporal dimension. The GCN-LSTM model integrates the advantages of the GCN and LSTM models, and this will significantly improve the accuracy of SST prediction.

### 2.7. Evaluation Solution of SST Prediction Models

This paper will use five models with different spatial correlation mining to predict SSTs. The first model is that of the LSTM based on the regular boundary division, and it includes three models based on different division methods—square division (LSTM-S), horizontal rectangle division (LSTM-H), and vertical rectangle division (LSTM-V)—to verify the optimal division method for SST prediction. The second model is ConvLSTM, which is based on convolutional sliding translation, to analyze the advantages of convolutional

sliding translation compared with regular boundary division. The third model is a model based on the clustering neural network (SOM-LSTM), which combines SOM and LSTM to analyze the advantages of the clustering neural network compared with convolutional sliding translation. Wei et al. [45] also used SOM-LSTM to achieve better SST prediction. The fourth model is the deep learning model (GCN-LSTM) based on GCN and LSTM, which is also the proposed model in this paper. This model realizes spatial correlation mining through a graph convolutional neural network. The fifth model is a conventional LSTM model (LSTM) without any spatial correlation mining. Next, we will use experiments to verify the advantages and effectiveness of the GCN-LSTM model compared with the other four models.

*2.8. Data Sets*

The data used in the experiments are the SST data for the East China Sea, which comes from the National Oceanic and Atmospheric Administration (NOAA) dataset of the U.S. Physical Sciences Laboratory (PSL). The laboratory provides high-resolution global daily average SST data. The selected spatial scope of sea area in experiments is the East China Sea. The latitude and longitude ranges are 21.125° N–30.875° N, 122.375° E–132.125° E, and the spatial resolution is 0.25° × 0.25°, as shown in Figure 4. The dataset time range is from 2010 to 2018, and the temporal resolution is in days.

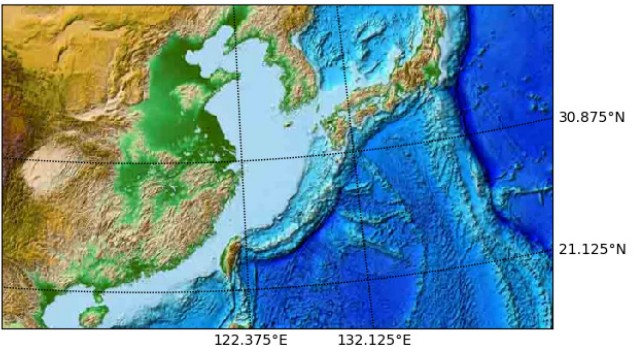

**Figure 4.** The spatial scope of the sea area for experiments.

## 3. Results

*3.1. Model Configuration and Evaluation Criteria*

In order to achieve better prediction results, the experiment uses hyperbands to identify the optimal hyper parameters. For LSTM-H, LSTM-S, and LSTM-V, the number of hidden units is 256, and the batch size is 64. The hyper parameters of the regular LSTM model are the same as for the above three models. The number of convolution kernels for the ConvLSTM model is 256, and the batch size is the same as for the LSTM model. For the SOM-LSTM model, the SOM size is (5, 5), the neighbor radius is set to 2, the nearest neighbor function is Gaussian, and the LSTM part is the same as that for the regular LSTM model. For the GCN-LSTM model, the LSTM part is the same as for the above LSTM model. In addition, for all models, the optimizer algorithm is Adam, the learning rate is 0.001, the number of training epochs is 1000, the early stopping mechanism is enabled, and the number of epochs for early stopping is 10. The detailed parameter settings for all the models are shown in Table 1.

**Table 1.** The detailed parameter information of the five different prediction models.

| Parameters | LSTM-H/S/V | ConvLSTM | SOM-LSTM | LSTM | GCN-LSTM |
|---|---|---|---|---|---|
| Time Step | | | 10 | | |
| Input Shape | (10, 16) | (10, 1600, 1, 1) | (10, 1600) | (10, 1600) | (10, 1600) |
| No. of LSTM Units | | | 256 | | |
| Size of Convolution Kernel | / | (5, 1) | / | / | / |
| Size of Convolution Step | / | (5, 1) | / | / | / |
| No. of Convolution Kernels | / | 256 | / | / | / |
| Batch Size | | | 64 | | |
| No. of Spatial Group | 100 | / | / | / | / |
| Spatial Scope | | 21.125° N–30.875° N | 122.375° E–132.125° E | | |
| Time Range—Training | | 1 January 2010 to 18 April 2016 | | | |
| Time Range—Testing | | 19 April 2016 to 31 December 2018 | | | |

In the experiment, the prediction performance of the model will be validated using the following evaluation criteria: root mean squared error (RMSE), mean absolute error (MAE), correlation coefficient (*r*), and mean absolute percentage error (MAPE). The specific definitions are as follows.

The RMSE is used to measure the deviation of the predicted value from the observed value. The smaller the value, the better the performance of the prediction model. The RMSE results are on the same level as the data, and they are more sensitive to particularly large or small errors, which can reflect the accuracy of the prediction model well. The equation for the RMSE is Equation (14).

$$\text{RMSE} = \sqrt{\frac{\sum_{i=1}^{n}(y_i - x_i)^2}{n}} \tag{14}$$

The MAE is the average of the absolute errors between the predicted and observed values, and it can effectively avoid the situation where errors between the predicted and values cancel each other out; thus, it accurately reflects the actual situation of errors between the predicted and observed values. Like for the RMSE, a smaller value indicates a better performance for the model. The MAE is less sensitive than the RMSE to very large or very small errors. The calculation equation for the MAE is Equation (15).

$$\text{MAE} = \frac{1}{n}\sum_{i=1}^{n}|y_i - x_i| \tag{15}$$

The *r* value is the measure for the degree of linear correlation between predicted and observed values, and it is often used to verify the validity of predicted values in deep learning models. Contrary to the RMSE and MAE, *r* is inversely proportional to the performance of the model, and the closer it is to 1, the stronger the correlation between the predicted and observed values. The equation for calculating *r* is Equation (11) in Section 2.5.

The MAPE is the average of the percentage error between the predicted and the actual values, which can effectively avoid the impact caused by a wide range of data. It is defined by Equation (16).

$$\text{MAPE} = \frac{1}{n}\sum_{i=1}^{n}\left|\frac{y_i - x_i}{x_i}\right| \times 100\% \tag{16}$$

*3.2. Effect Analysis for Regular Boundary Division*

As mentioned in Section 2.1, there are three different division methods: horizontal rectangular division, square division, and vertical rectangular division. The corresponding deep learning models for them are LSTM-H, LSTM-S, and LSTM-V respectively. In the sea

area selected by the experiment, there are a total of 1600 spatial points, and if each group is set to contain 16 points, there will be 100 groups. The division results for the three different methods are shown in Figure 5.

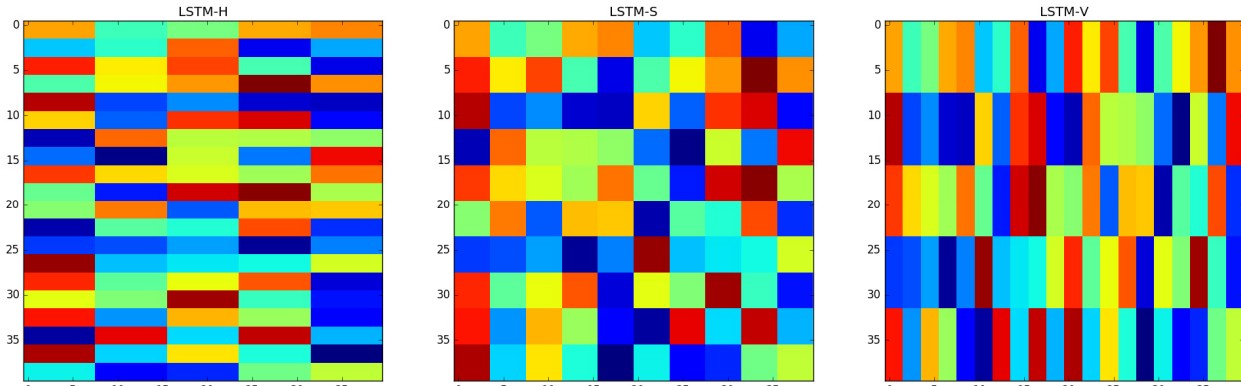

**Figure 5.** The division results of the sea area by three different division methods.

The three models with three different division methods are used to train and predict the SST in the selected sea area, respectively, and to generate the predicted SST in the entire test time range. Through the evaluation criteria, we can verify the differences in the influence from different division methods on the prediction model of the SST. At the same time, for the entire sea area without division, or for the entire sea area as a group, the LSTM model is used for SST training and prediction in order to compare the effect difference between division and non-division.

Through the comparison between the predicted values and observed values, the evaluation criteria of the four different models are shown in Table 2.

**Table 2.** The predictive effect of different division methods.

| Evaluation Criteria | LSTM | LSTM-H | LSTM-S | LSTM-V |
|:---:|:---:|:---:|:---:|:---:|
| MAE | 0.7108 | 0.3621 | 0.3505 | 0.3684 |
| RMSE | 0.8717 | 0.4691 | 0.4563 | 0.4767 |
| MAPE | 0.0287 | 0.0145 | 0.0140 | 0.0147 |
| $r$ | 0.9865 | 0.9937 | 0.9940 | 0.9935 |

It can be clearly seen from Table 2 that the spatial correlation implemented by regular boundary division improves the prediction accuracy for all evaluation criteria. At the same time, among the three different division methods, the predictive effect of LSTM-H is better than that of LSTM-V, and LSTM-S is the best model. This is also consistent with the law of SST in the actual situation. Latitude has a greater impact on temperature than longitude. Compared with vertical division, the latitude changes less, so the difference in SST within a group is smaller, and the law is more similar. The square division is a method between horizontal and vertical division. The SST changes in the dimensions of longitude and latitude are moderate, and the farthest distance between spatial points is smaller than in the horizontal and vertical methods, so it has the best prediction accuracy.

### 3.3. Effect Analysis for Spatial Feature Extraction by the Clustering Neural Network

In Section 3.2, experiments prove the good SST prediction ability from regular boundary division. The clustering neural network does not divide the sea area by regular shape and distance. The spatial correlation between the spatial points is mined and discovered through the SOM neural network. Based on that, the clustering of similar points is implemented. Next, we will verify the effect of the clustering neural network for SST prediction by experiments.

The above experiments prove that LSTM-S has the best prediction ability using the regular division method. The following experiment will use the SOM neural network to cluster the spatial points according to the time series of SST at each spatial point, and the clustering results will be input into the LSTM model to build a SOM-LSTM model by combining SOM and LSTM. Then, the SOM-LSTM model will be used for SST training and prediction. The comparison results of the SOM-LSTM and LSTM-S models after the training and prediction are shown in Table 3.

**Table 3.** The predictive effect of the SOM-LSTM and LSTM-S models.

| Evaluation Criteria | LSTM-S | SOM-LSTM |
|:---:|:---:|:---:|
| MAE | 0.3505 | 0.2991 |
| RMSE | 0.4563 | 0.3949 |
| MAPE | 0.0140 | 0.0122 |
| $r$ | 0.9940 | 0.9956 |

It can be clearly seen from Table 3 that the experimental results are consistent with the theory, and for each evaluation criterion, the SOM-LSTM model is better than the LSTM-S model. Therefore, the experiment proves that the clustering neural network breaks through the limitations of the regular boundary division, and as long as there is a potential similarity law, the spatial points that are relatively far away can also be clustered into a group, which further enhances the effect of spatial correlation and improves the SST prediction.

### 3.4. Effect Analysis of the Different Graphs for the Graph Convolutional Neural Network

Although the experimental results show that the clustering neural network further improves the SST predictive effect, as mentioned in Section 2.4, the clustering neural network still has explicit group boundaries, and the connections between nodes in different groups are ignored to some extent. Therefore, by combining the advantages of the convolution neural network and the clustering neural network, the spatial correlation implemented by the graph convolutional neural network (GCN) is introduced, and a spatiotemporal fusion prediction model, GCN-LSTM, based on the GCN and LSTM models, is proposed and designed as shown in Section 2.6. Next, the predictive effect of the GCN-LSTM model will be verified by experiments.

In the experiment, the threshold of correlation coefficient $r$ is used to generate the adjacency matrix of the graph structure. The selection of the threshold will directly affect the number of non-zero values of the adjacency matrix, and different thresholds will generate different adjacency matrixes. For the graph structure, when the $r$ value between two points is greater than or equal to the threshold, there is an edge between these two points. In order to verify the predictive effect of the SST prediction model more comprehensively, 11 different thresholds are selected. The $r$ values of the entire sea area and the adjacency matrixes corresponding to the 11 thresholds are shown in Figure 6. The first subgraph is the value of $r$ between any two points. The remaining 11 subgraphs are the adjacency matrixes for all thresholds. If there is an edge between two points, the corresponding position on the adjacency matrix diagram is red; otherwise, it is blue. So, the larger the red area, the greater the number of edges in the graph.

It can be clearly seen from Figure 6 that the $r$ values are distributed symmetrically along the diagonal line from the top left to the bottom right corner, which is consistent with the undirected graph structure of the SST data. The closer to the diagonal line, the higher the $r$ value. As the threshold increases, the red area where there are edges gradually becomes smaller and narrower, and this indicates that the number of edges in the graph structure gradually decreases. When the threshold is equal to 0.88, there are edges between most of the two points, and when the threshold is equal to 0.98, only the points with very strong correlation have edges, so the number of edges in the graph is very small. Through these experiments, we can generate the number of edges for the SST graph structure of

the selected sea area based on the 11 different thresholds. The number of edges for each threshold is shown in Table 4.

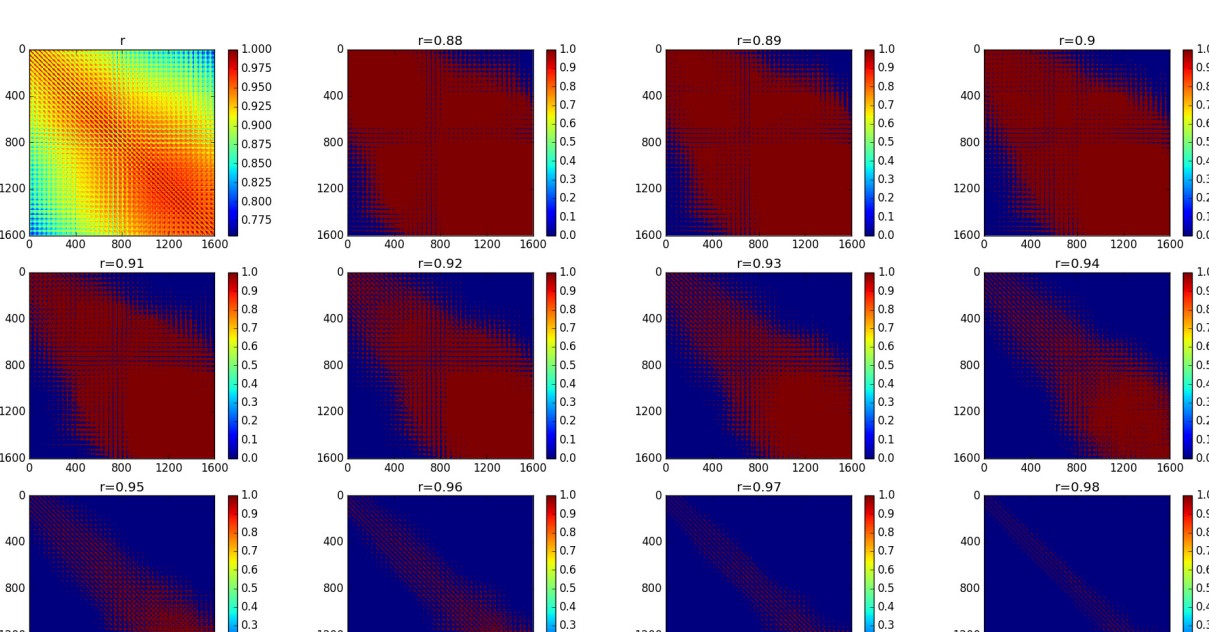

**Figure 6.** Adjacency matrixes for different thresholds.

**Table 4.** The number of edges for the different thresholds.

| Threshold | 0.88 | 0.89 | 0.90 | 0.91 | 0.92 | 0.93 |
|---|---|---|---|---|---|---|
| No. of edges | 2,216,234 | 2,078,242 | 1,897,754 | 1,669,650 | 1,402,046 | 1,133,382 |
| Threshold | 0.94 | 0.95 | 0.96 | 0.97 | 0.98 | / |
| No. of edges | 857,960 | 561,184 | 296,536 | 149,712 | 75,488 | / |

For different thresholds, the adjacency matrix of the graph is constructed by experimentation, and then the graph structure of the SST is generated. Each SST graph is input into the GCN-LSTM spatiotemporal fusion model for training and prediction. Then, the respective predicted SST values of the entire sea area within the test time range are generated, and the four evaluation criteria of the model are calculated through the predicted SST values and actual values, so the predictive effect of the GCN-LSTM model can be compared with itself when different thresholds are selected. The comparison results are shown in Figure 7.

It can be clearly seen from Figure 7 that the MAE, RMSE, and MAPE first decrease and then go up with the increase in the threshold, so the predictive effect of the model first improves and then deteriorates from the perspective of these three evaluation criteria. On the contrary, the correlation coefficient *r* first increases and then decreases, and it also indicates that the predictive effect of the model first improves and then deteriorates. This is consistent with the actual situation of the graph structure. When the threshold is small, an edge with an insufficient correlation relationship will be introduced, which will affect the predictive effect of the model. As the threshold increases, the edge with a weak correlation relationship is excluded, so the predictive effect of the model gradually becomes better. However, when the threshold continues to increase, only the edge with a strong correlation relationship exists. During this process, some valuable edges are excluded at the same time, resulting in too small a number of edges, which further affects the predictive effect of the model. Therefore, the predictive effect of the model deteriorates with the continuous increase in the threshold. It can be seen from the experiment that when the threshold is set to 0.92, all the evaluation criteria

are the optimal, and the model has the best predictive effect. So, in the following experiments, if it is not explicitly stated, the default threshold will be 0.92.

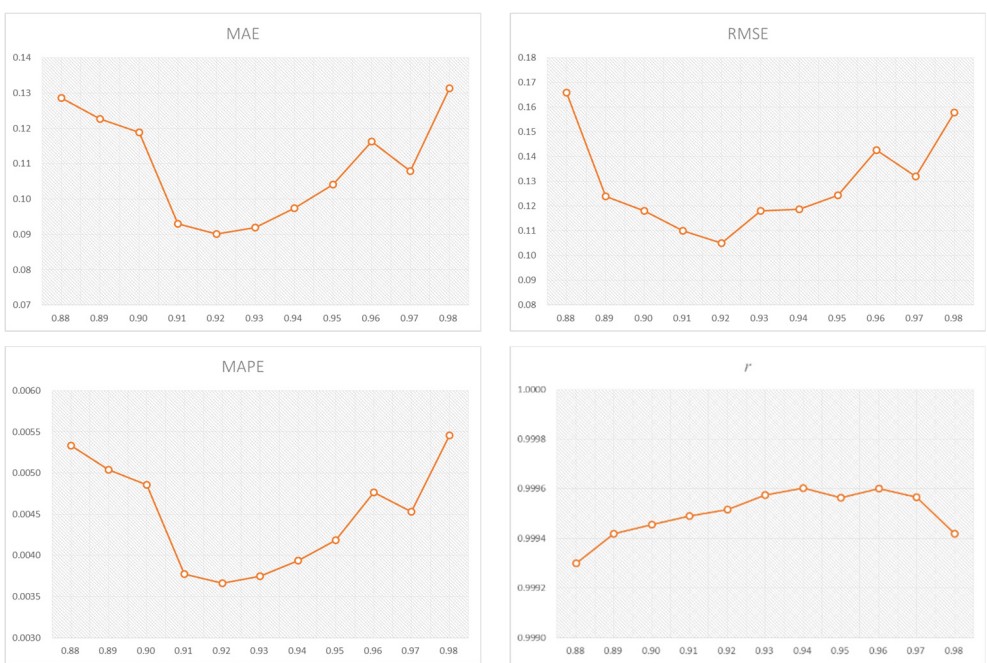

**Figure 7.** The predictive effect of the GCN-LSTM model with different thresholds.

### 3.5. Effect Analysis of the GCN-LSTM Model

In the sea area selected by the experiment, three spatial points are selected for model verification. The first one is at the boundary of the sea area and its coordinates of latitude and longitude are (124.625° E, 21.125° N). The other two are within the sea area. Their coordinates of latitude and longitude are (125.125° E, 27.125° N) and (129.875° E, 29.125° N), respectively. These are shown in Figure 8.

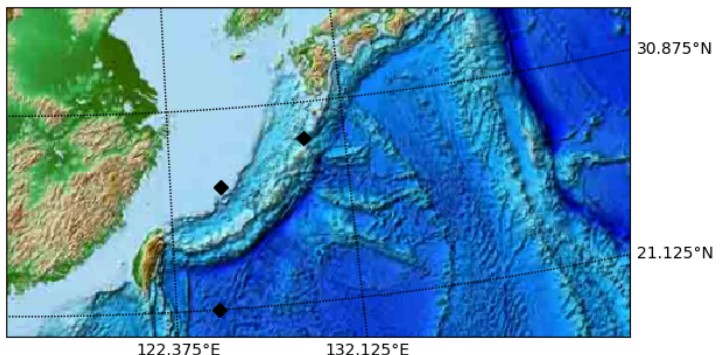

**Figure 8.** The selected three spatial points for SST prediction.

After the prediction of SST is completed, the comparison of the predicted SST and observed SST at the three selected spatial points of each model is shown in Figure 9. There are three figures. Figure 9A shows the comparison results for (124.625° E, 21.125° N). Figure 9B shows the comparison results for (125.125° E, 27.125° N), and Figure 9C shows the comparison results for (129.875° E, 29.125° N). In addition, each figure includes two subfigures, where subfigure (a) is the comparison for the entire test time range from 19 April 2016 to 31 December 2018. In order to more clearly compare the predictive effects between different models, a month is selected from 30 November 2017 to 30 December 2017 to compare the predictive effects between different models. The subfigure (b) shows the comparison results over a period of this month.

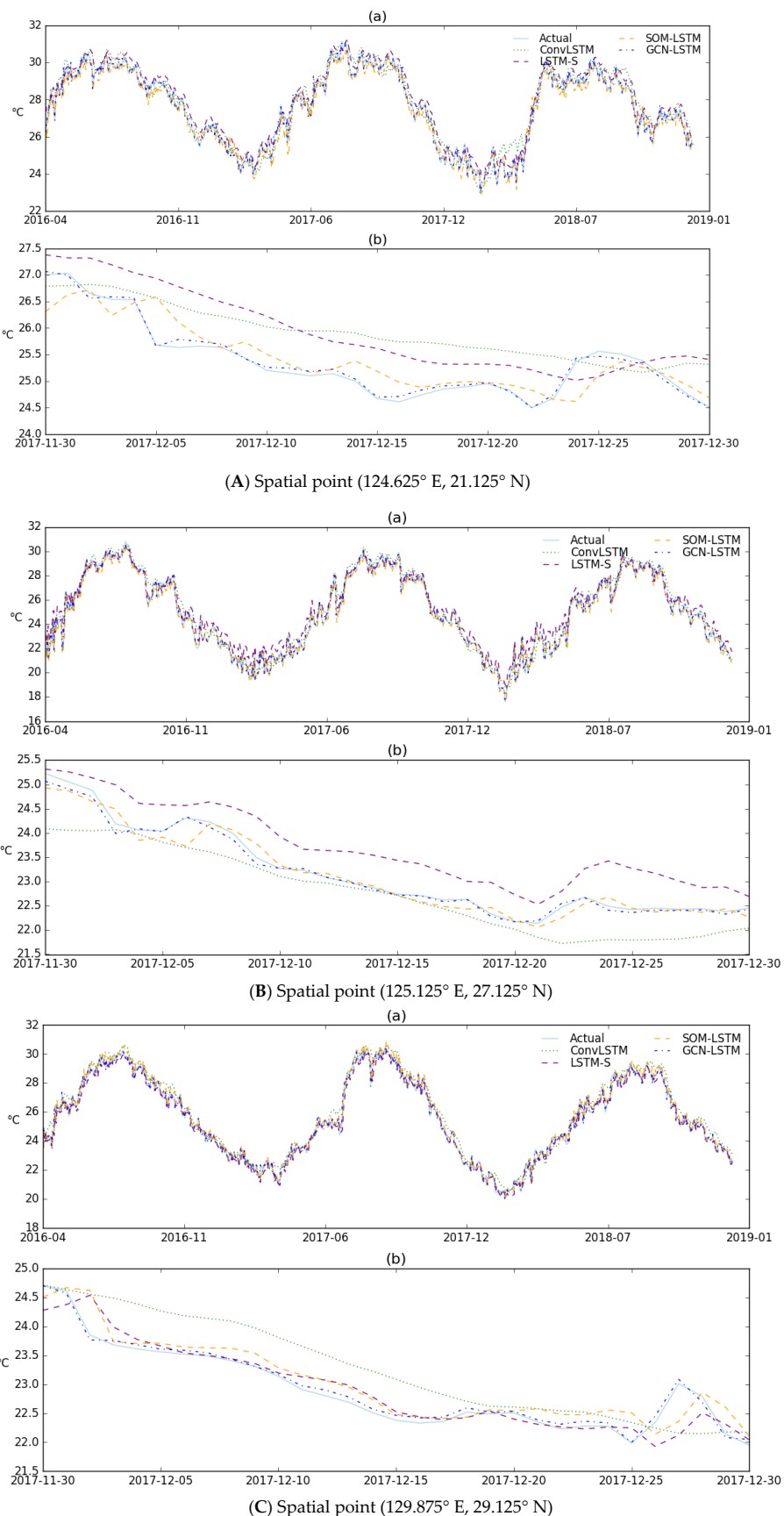

**Figure 9.** The comparison of different models at three different spatial points.

For the predictive effect of the different models with different spatial correlation mining methods, it can be clearly seen from the comparison figure for three spatial points that the ConvLSTM model with convolutional sliding translation has a similar effect to the LSTM-S model with the regular boundary division. SOM-LSTM with the clustering neural network is better than the ConvLSTM and LSTM-S models. The GCN-LSTM model, which uses the graph convolutional neural network to achieve spatial correlation, further improves the effect of SST prediction, and the trend of the predicted SST for all three spatial points is consistent with the trend of the observed SST.

The predictive effect of the model can be reflected more comprehensively through the evaluation criteria. The comparison results of the evaluation criteria for the four models used in the experiments at three spatial points are shown in Table 5.

**Table 5.** Comparison of evaluation criteria for different models at three spatial points.

| Spatial Point | Evaluation Criteria | ConvLSTM | LSTM-S | SOM-LSTM | GCN-LSTM |
|---|---|---|---|---|---|
| (124.625° E, 21.125° N) | MAE | 0.4532 | 0.4499 | 0.2888 | 0.0659 |
| | RMSE | 0.5659 | 0.5356 | 0.3643 | 0.0860 |
| | MAPE | 0.0168 | 0.0165 | 0.0105 | 0.0024 |
| | $r$ | 0.9833 | 0.9908 | 0.9930 | 0.9995 |
| (125.125° E, 27.125° N) | MAE | 0.5401 | 0.3176 | 0.3180 | 0.1071 |
| | RMSE | 0.7017 | 0.4431 | 0.4387 | 0.1407 |
| | MAPE | 0.0225 | 0.0129 | 0.0128 | 0.0044 |
| | $r$ | 0.9894 | 0.9940 | 0.9957 | 0.9995 |
| (129.875° E, 29.125° N) | MAE | 0.4432 | 0.2715 | 0.2526 | 0.0787 |
| | RMSE | 0.5568 | 0.3547 | 0.3373 | 0.1008 |
| | MAPE | 0.0174 | 0.0106 | 0.0098 | 0.0030 |
| | $r$ | 0.9927 | 0.9963 | 0.9970 | 0.9997 |

As we can see from Table 5, the values of the four evaluation criteria for the ConvLSTM model are all the worst. LSTM-S is similar to ConvLSTM and slightly better than the ConvLSTM model in four aspects, while SOM-LSTM using the clustering neural network is comprehensively superior to LSTM-S. Finally, GCN-LSTM greatly improves the predictive effect from the perspective of the four evaluation criteria and is the best model.

It can be concluded from the experiments that regular boundary division is slightly better than convolutional sliding translation for the predictive effect of a single point. The clustering neural network breaks through the limitations of regular division and improves the predictive effect compared with the above two methods. The spatial correlation achieved through the graph convolutional neural network takes into account the spatial correlation relationship of spatial points within and between groups and greatly improves the predictive effect in SST prediction.

Next, the predictive effect of different models in the entire sea area will be verified, and the advantages of the GCN-LSTM model proposed and designed in this paper will be also proved. When all the models complete the training and prediction of SST for the entire sea area and the predicted SST will be compared with the observed SST; the closer these two values are, the better the predictive effect of the model is. First, a day is selected to test the predictive effect of the different models. The day selected is 24 February 2018, and the comparison results of all models for this day are shown in Figure 10.

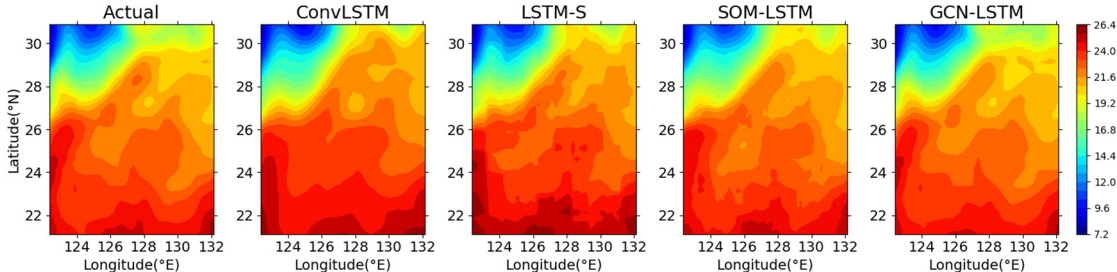

**Figure 10.** The comparison between the predicted and actual SST by the different models for the same day.

It can be seen from Figure 10 that the predicted results of each model are consistent with the change rule of SST, i.e., the higher the latitude, the lower the temperature. Furthermore, it can be seen from the figure that there are many differences between the predicted values and the actual values for the ConvLSTM model. Compared with the ConvLSTM model, LSTM-S makes many improvements, and the overall SST distribution is closer to the actual situation. The SOM-LSTM model has further improvement, and the predicted SST distribution is closer to the actual distribution than that predicted by LSTM-S and ConvLSTM. Finally, the prediction results of the GCN-LSTM model are only slightly different from the actual SST, which indicates a greatly improved SST predictive effect.

Through the above experiment, the SST prediction ability of different models has been verified by the single-day SST prediction, and the optimality of the GCN-LSTM model has been proved. If the entire test time range is taken into account, the predictive effect of the different models can be more fully reflected. This purpose can be achieved by calculating the average SST over the entire test time range for each spatial point in the selected sea area. Therefore, after the training and prediction are completed, the average predicted SST and the average actual SST at each spatial point can be obtained, and the comparison results of the predictive effect for different models can be realized in the entire sea area in the spatial dimension, as well as for the entire test time range in the temporal dimension. The comparison results are shown in Figure 11.

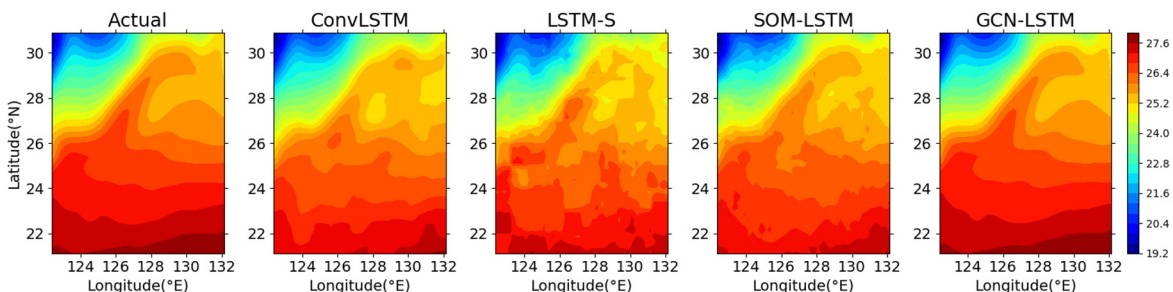

**Figure 11.** The comparison between the average predicted and the actual SST for the entire sea area.

It can be clearly seen from Figure 11 that the comparison results between the predicted and actual SST in the entire test time range are consistent with the comparison results of the single day. The spatial correlation implemented by the GCN and the GCN-LSTM models proposed in this paper achieves the best prediction effect, and the predicted value is almost equal to the actual value. This result also fully reflects the influence of different spatial correlation mining methods on the SST deep learning model. The clustering neural network is better than the regular boundary division and convolutional sliding translation methods, and the GCN is superior to the clustering neural network.

Next, different models will be compared by the evaluation criteria, MAE and MAPE. The first is the comparison result for the MAE. By calculating the MAE of 1600 spatial points in the sea area, MAE figures corresponding to different models can be generated, and then MAE figures of different models can be used to reflect whether the prediction effect is good or not.

The comparison results of the MAE figures for the different models are shown in Figure 12. The 1600 spatial points in the figure are arranged in the form of $40 \times 40$ in the latitude and longitude dimensions, and the color bars in the figure are used to indicate the quality of the prediction results. The closer the color is to blue, the better the predictive effect is, and the closer the color is to red, the worse the predictive effect is.

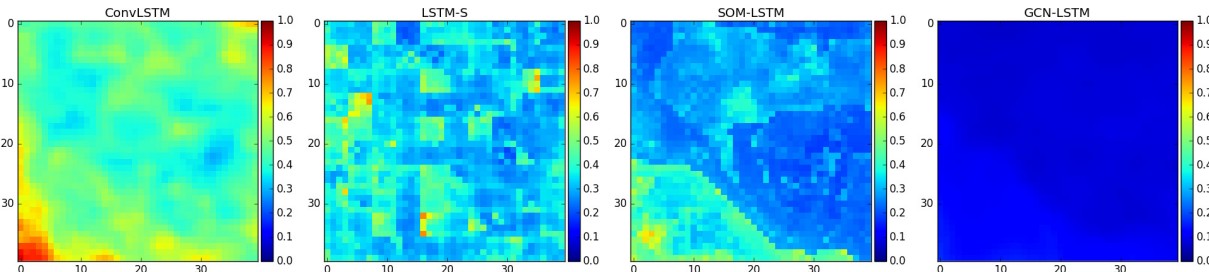

**Figure 12.** MAE comparison for the different models.

As we can see from Figure 12, the predictive effect at most spatial points using ConvLSTM is not ideal, with less blue, more yellow, and a small amount of red. The red spatial points of the LSTM-S model have almost disappeared, while most of the spatial points are blue, and there are still some yellow spatial points. When switching to the SOM-LSTM model with the clustering neural network, most of the spatial points are blue or light blue, and there are some light-yellow spatial points in the lower left corner. The GCN-LSTM model using the graph convolutional neural network is the purest, and the entire sea area is dark blue, which fully reflects the optimality and stability of the predictive effect of this model.

Next is the evaluation criterion MAPE, which reflects the relative error between the predicted value and the actual value and indicates the predictive effect of the model through relative measurement. The MAPE comparison results for the different models for $40 \times 40$ spatial points in the entire sea area are shown in Figure 13.

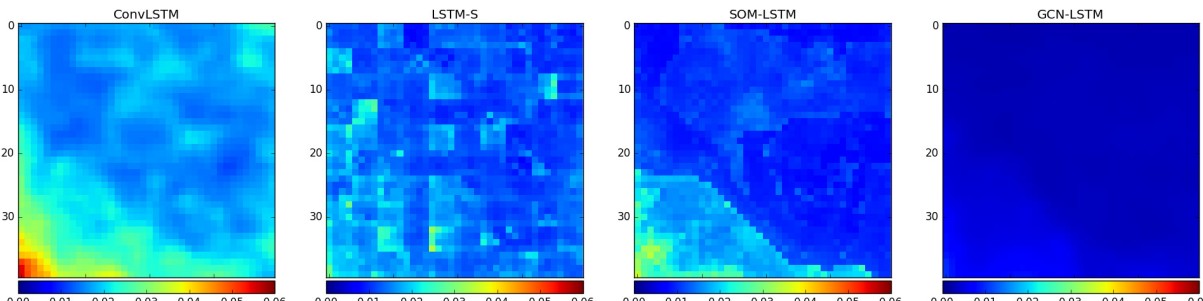

**Figure 13.** MAPE comparison for the different models.

Because the value of MAPE is usually much smaller than the value of the MAE and RMSE, the value range of the color bar for the MAPE is much smaller than that for the MAE and the RMSE, so we only need to select the value range from 0 to 0.06. The MAPE of all the models is also within this range. As we can see from Figure 13, the comparison results of the MAPE are also consistent with those for the MAE and the RMSE. The ConvLSTM, LSTM-S, and SOM-LSTM models show mainly light blue and blue, with some yellow points, and ConvLSTM also has a few red points. GCN-LSTM behaves as usual in that the entire sea area is dark blue. Therefore, the advantages and stability of the spatial correlation through the GCN-LSTM model are proved once again through the MAPE comparison.

Finally, the effect of the different spatial correlation mining methods and different models for SST prediction is comprehensively compared through the four evaluation criteria. The comparison results of the evaluation criteria for the different models are shown in Table 6.

**Table 6.** Comparison of the average MAE, RMSE, MAPE, and *r* for the different models.

| Evaluation Criteria | ConvLSTM | LSTM-S | SOM-LSTM | GCN-LSTM |
|:---:|:---:|:---:|:---:|:---:|
| MAE | 0.4670 | 0.3506 | 0.2991 | 0.0901 |
| RMSE | 0.6047 | 0.4564 | 0.3949 | 0.1188 |
| MAPE | 0.0191 | 0.0140 | 0.0122 | 0.0036 |
| *r* | 0.9898 | 0.9941 | 0.9956 | 0.9996 |

It can be seen from Table 6 that the average evaluation criteria are also completely consistent with the above experimental results. The results for the ConvLSTM and LSTM-S models are close to each other, while those for the LSTM-S are a little bit better. After the introduction of the clustering neural network, the SOM-LSTM improves the predictive effect with respect to the MAE, RMSE, MAPE by nearly 40% compared with the previous two models, and the correlation coefficient *r* also becomes bigger. The MAE, RMSE, and MAPE of the GCN-LSTM model using the graph convolutional neural network decrease by 69% compared with the previous three models, and it also has the biggest correlation coefficient *r*. The experimental results prove that the spatial correlation implemented by the graph convolutional neural network and the GCN-LSTM model proposed in this paper are effective, optimal, and stable for SST prediction.

## 4. Discussion

In this paper, we explored the spatial correlation discovery and mining methods and proposed the spatiotemporal fusion model for SST prediction. The first method uses regular boundary division [43], including horizontal rectangular division, vertical rectangular division, and square division. It can improve the predictive effect to a certain extent compared with the model without spatial correlation; however, it depends on the regular Euclidian space and requires explicit division. So, we introduced the second method, which consists of convolutional sliding translation using the convolutional neutral network [33]. It does not require explicit division and still depends on the regular Euclidian space. The third method solves this problem using the clustering neural network for spatial feature extraction [45]. However, it introduces a new problem, that is, the lack of spatial correlation around the boundary of groups. Finally, we proposed the spatial correlation mining method using the graph convolutional neural network and SST graph structure. The proposed method can solve the above problems and more effectively mine the spatial correlation of SSTs. Based on the proposed method, we also designed the GCN-LSTM spatiotemporal fusion model for SST prediction. It combines the spatial advantages of the GCN and the temporal advantages of LSTM and greatly improves the SST prediction.

We designed five models to verify the above methods and the proposed GCN-LSTM model. Through various experiments, the method of regular boundary division was found to have a similar predictive effect to the method using the convolutional neural network. The clustering neural network method achieves much better prediction accuracy than the first two methods. The methods using the graph convolutional neural network and the GCN-LSTM model further improve the accuracy and have the best predictive effect. A more accurate SST prediction will provide solid technical support for responding to the challenges in marine biology [1,2], the global climate [3,4], and extreme weather events [5].

## 5. Conclusions

The SST data on the East China Sea were selected, and five models were verified by various experiments. By comparison with other methods and models, we prove that the proposed spatial correlation mining method using the graph convolutional neural network and the proposed GCN-LSTM model can effectively capture the SST features from both the spatial and temporal dimensions and achieve better accuracy for SST prediction. The results show an RMSE of 0.1188 °C and an MAE of 0.0901 °C. The MAPE is 0.0036, and the

*r* reaches up to 0.9996. The proposed method and model can solve the spatial prediction problem and improve the effect of SST prediction.

**Author Contributions:** Conceptualization, J.L. and L.W.; methodology, J.L.; software, F.H. and P.X.; validation, J.L., L.W. and D.Z.; formal analysis, J.L.; investigation, J.L.; resources, J.L.; data curation, L.W.; writing—original draft preparation, J.L.; writing—review and editing, L.W., F.H., P.X. and D.Z.; visualization, J.L.; supervision, D.Z.; project administration, D.Z.; and funding acquisition, J.L. and D.Z. All authors have read and agreed to the published version of the manuscript.

**Funding:** This paper was funded by the National Key R&D Program of China (2016YFC1401900), the National Natural Science Foundation of China (Grant No. 82011530399), the Zhejiang Province Key Research and Development Program (Grant No. 2021C01189), the Leading Talents of Science and Technology Innovation in Zhejiang Province (Grant No. 2020R52042), the Science and Technology Department of Zhejiang Province (Grant No. LGG21F020007), and the Major Scientific Research Innovation (team) Project (Grant No. 2021XZ015).

**Data Availability Statement:** The data that support the findings of this study are openly available from the NOAA at ftp://ftp.cdc.noaa.gov/Datasets/noaa.oisst.v2.highres.

**Acknowledgments:** We are grateful to the National Oceanic and Atmospheric Administration (NOAA) for the SST data used in this study.

**Conflicts of Interest:** The authors declare no conflicts of interest.

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
