# Peer review of "Spatiotemporal Fusion Prediction of Sea Surface Temperatures Based on the Graph Convolutional Neural and Long Short-Term Memory Networks"

_water, doi:10.3390/w16121725_

Round 1

Reviewer 1 Report

Comments and Suggestions for Authors

Sea surface temperature is a key parameter in characterizing global climate change and have significantly impact on marine biology. This paper realizes SST prediction by exploring spatial correlation mining methods and extracting SST features in spatial and temporal latitude by combining with Long Short-Term Memory Network (LSTM). The research is of significance and the experimental process of the paper is complete. Compared with the previous related SST prediction models, the accuracy of the model proposed in the paper is improved, which is of positive significance to the development of SST prediction.

But the overall structure of the paper, as well as the use of words and grammar need to be improved.

The major comments are as follows:

1.       In the introduction section, the research background of the SST prediction problem is not presented in a complete manner. As far as I know, the LSTM-AdaBoost model has also been used for SST prediction and also achieved desirable results.

2.       The discussion section of the article simply describes the advantages and disadvantages of the previous methods as well as the advantages of the proposed method, but there is no discussion of the applications and implications of the research. References citation was needed in this part.

3.       In this article, the parameter 'r' has two meanings; 'r' is both the Pearson's correlation coefficient and the threshold, which can be easily misinterpreted.

4.       Section 3.1 and 3.2 should be put in section 2. They are not results.  

5.       A flowchart of the paper is recommended.

The minor comments are follows:

1.       Line 11, Marine related fieldsàMarine-related fields。

2.       Line 28, have significantly impact toàhave a significant impact on。

3.       Line 33, “so it limits the development and accuracy for SST prediction” à“which limits the development and accuracy of SST prediction”。

4.       Line 46, haveàhas。

Comments on the Quality of English Language

There are some colloquial terms in the paper.

Reviewer 2 Report

Comments and Suggestions for Authors

The limitations of current Sea Surface Temperature (SST) prediction methods by enhancing spatial correlation utilization through a Graph Convolutional Neural Network (GCN). It combines GCN with Long Short-Term Memory (LSTM) networks to create a spatiotemporal fusion model (GCN-LSTM) for improved SST prediction accuracy. The proposed model effectively captures SST features in both spatial and temporal dimensions, and experimental results confirm its superior performance. following points may consider to improve the manuscript.

1.      Write examples of the significant impacts of sea surface temperature (SST) changes on marine biology, global climate, and extreme weather events.

2.      How do existing numerical prediction models for SST operate, and what are their specific limitations in replicating the physical evolution of SST?

3.      What advancements in deep learning technology have most significantly contributed to the improved accuracy of SST predictions?

4.      How do the different types of neural network models (e.g., feedforward, LSTM, GRU, CNN) specifically contribute to the accuracy and efficiency of SST prediction?

5.      What specific advantages do neural networks and deep learning technologies provide when there is a large amount of SST data available, especially with the development of remote sensing technology?

6.      How do the methods proposed by researchers like Choi et al., Usharani et al., and Zhang et al. differ in their approach to SST prediction, and what are the unique contributions of each?

7.      Can you explain in more detail how the graph convolutional network (GCN) addresses the limitations of traditional methods in dealing with the spatial correlation between marine spatial points?

8.      What specific improvements in SST prediction accuracy were observed with the regular boundary division for spatial interference elimination?

9.      How does the convolutional sliding translation method enhance spatial feature extraction compared to traditional methods?

10.  What are the optimal values for the convolutional kernel size (Kh, Kv) and step size (Sh, Sv) in the context of SST data?

11.  What advantages do cluster neural networks provide over regular boundary division and convolutional sliding translation for SST prediction?

12.  Can you elaborate on the role of SOM in spatial feature extraction and its effectiveness compared to other clustering methods?

13.  What criteria are used to determine the edges in the graph data structure, and how does this affect the performance of the GCN?

14.  What threshold value (α) for the correlation coefficient (r) is considered optimal for constructing the adjacency matrix?

15.  How does the normalization process in Equations (7) to (9) prevent issues like gradient explosion or disappearance?

16.  What is the impact of the prediction window size (F) on the performance of the GCN-LSTM model?

17.  How does the proposed GCN-LSTM model scale with an increase in the number of spatial points (P) and the number of days (D) in the SST time series data?

18.  What underlying patterns in SST data justify the observation that square division (LSTM-S) offers the best prediction accuracy compared to horizontal (LSTM-H) and vertical (LSTM-V) divisions? Are there specific geographical or oceanographic factors contributing to this outcome?

19.  Why do MAE, RMSE, and MAPE metrics initially decrease and then increase with higher r threshold values? What does this indicate about the optimal balance between including and excluding edges in the graph structure for accurate SST prediction

Comments on the Quality of English Language

 Extensive editing of English language required

Round 2

Reviewer 1 Report

Comments and Suggestions for Authors

My suggestions have been well considered and the paper has been improved.